caregiver; carer; depression; group intervention; India; mental health

**Corresponding author:**
Kaaren Mathias;
Email: kaaren.mathias@canterbury.ac.nz

C.R.B. and P.S.P. are joint first authors.

# Does the *Nae Umeed* group intervention improve mental health and social participation? A pre–post study in Uttarakhand, India

Christopher R. Bailie[1], Pooja S. Pillai[2], Atul Goodwin Singh[2], Jed Leishman[3], Nathan J. Grills[1] and Kaaren Mathias[2,4] 

[1]Nossal Institute for Global Health, University of Melbourne, Parkville, VIC, Australia; [2] Burans, Herbertpur Christian Hospital, Emmanuel Hospital Association, Dehradun, India; [3]Department of Medicine, University of Otago, Dunedin, New Zealand and [4]Faculty of Health, University of Canterbury, Christchurch, New Zealand

## Abstract

There are few evidence-based interventions to support caregiver mental health developed for low- and middle-income countries. *Nae Umeed* is a community-based group intervention developed with collaboratively with local community health workers in Uttarakhand, India primarily to promote mental wellbeing for caregivers and others. This pre–post study aimed to evaluate whether *Nae Umeed* improved mental health and social participation for people with mental distress, including caregivers. The intervention consisted of 14 structured group sessions facilitated by community health workers. Among 115 adult participants, 20% were caregivers and 80% were people with disability and other vulnerable community members; 62% had no formal education and 92% were female. Substantial and statistically significant improvements occurred in validated psychometric measures for mental health (12-Item General Health Questionnaire, Patient Health Questionnaire-9) and social participation (Participation Scale). Improvements occurred regardless of caregiver status. This intervention addressed mental health and social participation for marginalised groups that are typically without access to formal mental health care and findings suggest *Nae Umeed* improved mental health and social participation; however, a controlled community trial would be required to prove causation. Community-based group interventions are a promising approach to improving the mental health of vulnerable groups in South Asia.

## Impact statement

In low- and middle-income countries (LMICs) such as India, there are growing numbers of people with chronic illnesses, who are mostly cared for by their families. This caregiver role providing informal and regular care to someone with a long-term need for care is performed by one in six adults in LMIC. Caregivers are at increased risk of mental health problems such as depression and anxiety. Interventions developed in high-income settings to support caregiver mental health, include educational and counselling interventions, however few caregiver interventions have been developed and shown as effective in LMICs. This is important because interventions work best when they are designed to meet local needs and are sensitive to cultural, social and economic contexts. In India, caregivers who are less educated, financially worse-off, socially isolated and typically female generally have worse mental health. These same factors make getting help from health services more challenging, meaning interventions must also be delivered in ways that are accessible to those in need. In this study we evaluated the effectiveness of a locally developed group intervention, *Nae Umeed*, which aimed to promote mental health in Dehradun, Uttarakhand, India in informal urban parts of Dehradun through a community mental health non-profit in late 2020. Although the intervention was initially designed to support caregivers, participants also included people with disabilities and other vulnerable community members. Community health workers facilitated fourteen structured group sessions on topics such as self-care and accessing entitlements. We collected data on measures of mental health and social participation before and after the intervention. Mean scores on these measures improved significantly. These findings suggest *Nae Umeed* can improve the mental health of participants, and suggeststhat locally developed community-based group interventions can help to address mental health disparities in South Asia where there are few formal treatment services.

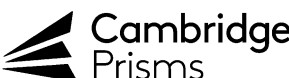

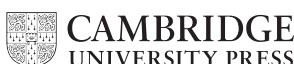

## Background

Caregiving is an increasingly significant global public health issue as increasing proportions of ageing populations live with disability (Crimmins et al., 2016). Issues around the well-being of caregivers are important for their personal health and the people they provide care for, as well as for the sustainability of health and social care systems to which they are integral (Talley and Crews, 2007). This latter consideration is especially relevant in countries like India, where the demographic transition towards greater noncommunicable disease burden is not matched by increases in health systems capacity (Bollyky et al., 2017), and where family members provide nearly all care for individuals with chronic illness or disability.

Caregiving is associated with both reward and fulfilment, as well as significant challenges (Schulz and Sherwood, 2008). These challenges, termed 'burden' (Platt, 1985), can adversely affect caregiver physical, mental and social well-being (Schulz and Sherwood, 2008). Negative mental health impacts from caregiving are consistently described and depend on local cultural and socioeconomic contexts (Bastawrous, 2013) in addition to individual and interpersonal factors, including the relationship between the caregiver and person with disability, type of disability, and age and gender of the caregiver (Pinquart and Sörensen, 2003).

Despite India's huge diversity, there are common contextual factors that likely shape caregivers' mental health. Non-biomedical explanatory models of mental illness are widespread (Poreddi et al., 2015; Chakrabarti, 2016), which typically ascribe responsibility for the illness to the person being cared for (Poreddi et al., 2015), and lead to societal stigma and social exclusion (Mathias et al., 2015a; Venkatesh et al., 2015). As in other parts of the world (Macintyre et al., 2018), economic disadvantage is strongly associated with mental ill-health (Mathias et al., 2015b), and in 2019 10% of the Indian population lived below the international poverty line of $2.15 USD/day (The World Bank, 2022). The gender relations in India mean that most caregivers are female (Janardhana et al., 2015; Chakrabarti, 2016). Women in India may experience greater challenges in sustaining caregiving due to systematic disadvantage, leading to feelings of hopelessness and overwhelming stress (Mathias et al., 2019; World Economic Forum, 2021). India's existing health system is not geared towards supporting caregivers' mental health (Chakrabarti, 2016) due workforce shortages, limited public mental health services and high out-of-pocket costs for consumers (Patel et al., 2015).

Caregiver 'burden' and associated mental health impacts in India have been described in those caring for people diagnosed with stroke (Mandowara et al., 2020), cancer (Menon et al., 2022), cirrhosis (Hareendran et al., 2020), psychosocial disability (Brinda et al., 2014; Stanley et al., 2017; Singh et al., 2021) and dementia (Pattanayak et al., 2010; Srivastava et al., 2016). However, a large proportion of those with disability who receive care do not have a formal diagnosis (Chakrabarti, 2016). Across different disabilities, commonly identified predictors of higher caregiver 'burden' or poorer mental health in these studies include female gender (Kumar and Gupta, 2014; Mandowara et al., 2020; Madavanakadu et al., 2022), social isolation (Jagannathan et al., 2014; Bapat and Shankar, 2021), economic disadvantage (Bapat and Shankar, 2021; Madavanakadu et al., 2022), fewer years of education (Jagannathan et al., 2014; Mandowara et al., 2020; Bapat and Shankar, 2021; Menon et al., 2022) and higher care-needs (Brinda et al., 2014; Mandowara et al., 2020). Finally, societal stigma operates towards caregivers of people with particular disabilities such as epilepsy (Bapat and Shankar, 2021) and psychosocial disability (Mathias et al., 2015a; Singh et al., 2016; Mathias et al., 2019; Dijkxhoorn et al., 2022), as well as towards the people they provide care for.

Although existing literature provides a strong rationale to intervene to address caregiver mental health in India, little evidence exists on how this should be done. Studies from high-income settings generally support the short-term effectiveness of non-pharmacologic interventions for improving the well-being and mental health of caregivers (Yesufu-Udechuku et al., 2015; Gabriel et al., 2020; Teahan et al., 2020; Lambert et al., 2021; Wiegelmann et al., 2021), including in group settings (Sörensen et al., 2002; Cheng and Zhang, 2020; Hovadick et al., 2021; McLoughlin, 2022). However, these studies are of variable quality, and methods for reporting interventions and assessing effectiveness are heterogeneous. Evidence from low- and middle-income countries (LMICs) is relatively scarce (Hinton et al., 2019; Gabriel et al., 2020). Within India, a variety of interventions have been trialled at small scale and with mixed results (Das et al., 2006; Dias et al., 2008; Kulhara et al., 2009; Chakraborty et al., 2014; Chatterjee et al., 2014; Lamech et al., 2020; Baruah et al., 2021; Singh et al., 2021; Sims et al., 2022; Stoner et al., 2022).

In this context, community-based group interventions offer several potential advantages. Community settings may be more accessible and acceptable than healthcare facilities, (Kohrt et al., 2018) and have been advocated as a specific low-resource strategy (Stanley et al., 2017). Groups also provide mechanisms for strengthening social and peer support (Hoddinott et al., 2010; Gailits et al., 2019; Morrison et al., 2019). On the other hand, group interventions may exacerbate existing inequalities with more educated participants engaging more effectively (Hoddinott et al., 2010). In India, several group interventions for caregivers have been implemented with reasonable feasibility and acceptability (Lamech et al., 2020; Sims et al., 2022; Stoner et al., 2022), although evidence for effectiveness is limited (Malini, 2015). There is an urgent need for cost-effective, equitable and sustainable interventions to strengthen caregiver mental health in LMICs.

*Nae Umeed* is a community-based group intervention that aims to improve mental health and social inclusion among disadvantaged caregivers of people with disability. The aim of this study is to assess the effectiveness of *Nae Umeed* in improving mental health and social participation among participants in Dehradun, Uttarakhand, India in 2020–2021, and to explore how effectiveness varies with socio-demographic identity.

## Methods

### Intervention

*Nae Umeed* was developed by Burans, a community-based partnership project administered by Herbertpur Christian Hospital seeking to improve mental health in communities of Uttarakhand (Burans, 2022). *Nae Umeed* was informed by previous research identifying women caregivers of people with disability as at high risk of social exclusion and strain (Mathias et al., 2019). It aims to build skills and knowledge in self-care, caregiving, psychosocial well-being, behaviour management, accessing support and entitlements, and management of household finances. The curriculum was developed collaboratively by community health workers, public health practitioners and mental health practitioners working in Uttarakhand in 2017. *Nae Umeed* was piloted with 15 groups of caregivers in 2019 and in response to feedback, additional content on household budget management and access to government entitlements were added.

In this study, *Nae Umeed* was delivered in a series of 14 group sessions, with five to seven participants per group. One-hour sessions were delivered weekly using a structured curriculum that covered topics linked to managing mental distress (modules 1–9) as well as managing household finances (modules 10–14). (Parinaam Foundation, 2014; Emmanuel Hospital Association, 2019). Recognising that most participants were not caregivers, facilitators adapted intervention content by providing examples that linked to experiences of psychosocial distress more broadly. Table 1 outlines the topics covered by the *Nae Umeed* module and they can be seen as relevant for people with mental distress. Participants were allocated into groups from their local community. Venues were chosen to maximise physical distancing and privacy. Sessions tools included visual aids from the manuals, whole group or small group discussions, role play activities, group team-building activities, and group revision quizzes. Several modules included short homework assignments, for example discussing an aspect of the session content with family members. Over the course of the intervention, participants were provided with several pamphlets related to the sessions, for example on self-care. The pamphlets on self-care were illustrated and the content was discussed with practical examples to cater to all levels of literacy.

Sessions were facilitated by nine community health workers, who facilitated two groups each (yielding a total of 18 groups), and also supported recruitment. Facilitators were trained to deliver *Nae Umeed* using a participatory facilitation style. Trainers were Burans project officers who had a minimum of 5 years working in community development and were qualified with a master's degree in social work. Due to the COVID-19 pandemic, facilitator training was provided online and supplemented with interactive discussions on each module using Whatsapp, as well as face-to-face meetings where possible.

Facilitators referred illness-specific queries about how to manage people with disability to the health professionals leading a disability programme at Herbertpur Christian Hospital, which hosted the implementation of *Nae Umeed*.

## Study design

The study design was an uncontrolled pre–post (before–after) study.

## Participants and setting

Given the real-world setting of this trial with high rates of mental distress post-lockdown, we elected to invite as many participants as community facilitators could accommodate in groups. Pragmatically they proposed they could manage up to a maximum of 18 groups with a maximum of 7 members per group, thus we invited a total of 126 people to participate in the intervention.

This intervention study was implemented from August to November 2020, when India was emerging from a harsh 12-week lockdown in the first year of the COVID-19 pandemic. There was widespread anxiety and reduced freedom of movement for most people. The setting was the urban and semi-urban slum areas of Dehradun. Burans staff invited individuals to participate in *Nae Umeed* through existing project networks involving people with disability and their household members. To be considered eligible to take part in the study, individuals had to be at least 18 years old, plan to reside in the area for the following 15 weeks and either be a caregiver or be a household member of a person with disability or identify themselves as experiencing significant psychosocial stress. In a setting with limited access to health care or social support for disability or mental health care, we used inclusive criteria and disability referred to any household member who had impaired function or ability to carry out activities of daily living. People with disabilities represented included people with sensory deficits, loco-motor challenges as well as psychosocial disability, although the majority of participants would not have had a formal mental health–related diagnosis or be receiving formal support or treatment (Mathias et al., 2015a). There was no requirement regarding the duration of caregiving or caregiving role (i.e. primary caregiver or other). Although the intervention was designed for caregivers, parameters for participation included other community members with mental distress to increase opportunities for social support (Gailits et al., 2019; Morrison et al., 2019) and reduce labelling and stigma of group members (Mathias et al., 2015a, 2019). In instances where participants included caregivers and people with disability from the same household or family, they participated in different groups. Recruitment was performed by Burans staff.

Consistent with the ethics approval, informed verbal consent was obtained and documented on forms by health workers who observed and signed that they had witnessed the consent process in line with recommended processes for meaningful informed consent (Bhutta, 2004).

## Outcomes

The primary outcomes were the Patient Health Questionnaire-9 (PHQ-9), which indicates risk of depression, and the short General Health Questionnaire (GHQ-12), which measures mental distress. The PHQ-9 has been validated in diverse settings in India and has shown stable performance across demographic subgroups and time (De Man et al., 2021). The GHQ-12 has been widely validated as a screening instrument for depression, including in India, and has been found to be robust across gender, age and educational level

**Table 1.** Summary of topics covered in the *Nae Umeed* group intervention by session

| Session | Topic summary |
|---------|---------------|
| 1. | Introduction to group and curriculum. Discussion of roles of caregivers |
| 2. | Mental illness: causes and symptoms |
| 3. | Importance of communication when caring for someone with mental illness |
| 4. | Techniques for behaviour modification |
| 5. | Medications: treatment plans, side effects |
| 6. | Effects of alcohol on health |
| 7. | Stress management techniques |
| 8. | Self-care |
| 9. | Recap session |
| 10. | Introduction to financial planning |
| 11. | Budgeting; tracking income and expenses |
| 12. | Strategies for saving money |
| 13. | Borrowing money safely |
| 14. | Recap of financial literacy session |

(Goldberg et al., 1997). The secondary outcome was change in score on the Participation Scale (P-scale) (van Brakel et al., 2006), which was designed to measure client-perceived social participation and developed and validated in South Asia.

### Data collection

Pre-intervention data were collected in the 2 weeks before starting the intervention. Post-intervention data were collected 3–4 months later, within 3 weeks of completion of the intervention. Demographic variables were recorded at both pre- and post-intervention outcome assessments. Data collection was performed by three Burans project officers (who were not involved as group facilitators), who recorded participants' verbal responses to questions. Data were checked by team leaders, and queries or inconsistencies clarified with team members or participants where necessary.

### Statistical analysis

Data analysis was performed using R version 4.1.2 (R Core Team, 2021). Participant ages were summarised as a median and range, and categorical demographic variables as counts and sample proportions. Participant demographic data recorded at the pre-intervention assessment were used for all analyses, except when this data was missing, in which case data recorded at the post-intervention assessment were used if available. Primary and secondary outcomes were assessed as mean score change among participants who completed both pre- and post-intervention assessments. Score change distributions for each outcome were visually inspected for normality using Q–Q plots. Confidence intervals (Cis) and p-values for paired two-sided t-tests were calculated using the *t.test* function. Due to higher than anticipated enrolment of non-caregivers, a post-hoc subgroup analysis of both primary and secondary outcomes by caregiver status (caregiver or non-caregiver) was performed to specifically investigate change among caregivers. As a further exploratory analysis, multivariable linear regression models were fit to change in each outcome score, including pre-intervention score and all demographic variables as predictors. Statistical significance was assessed at a threshold of $p = 0.05$ without adjustment for multiple comparisons.

### Trial registration

The study protocol was retrospectively registered with the Australia New Zealand Clinical Trials Registry (registration number: ACTRN12623000047695).

### Results

Recruitment was completed in early-to-mid August 2020. Overall, 124 people agreed to take part in the intervention. Eight subsequently left due either to migration or to conflicting employment commitments. The remaining 116 were recruited as study participants and completed pre-intervention data collection (Figure 1). Data from the 115 participants who completed follow-up in late November and early December 2020 were analysed.

The median age of participants was 35 years, 106 (92%) were female and 71 (62%) reported having completed no formal education (Table 2). There were 23 (20%) participants identifying as caregivers, 75 (65%) people with disability and 17 (15%) others (comprising other vulnerable community members identified by

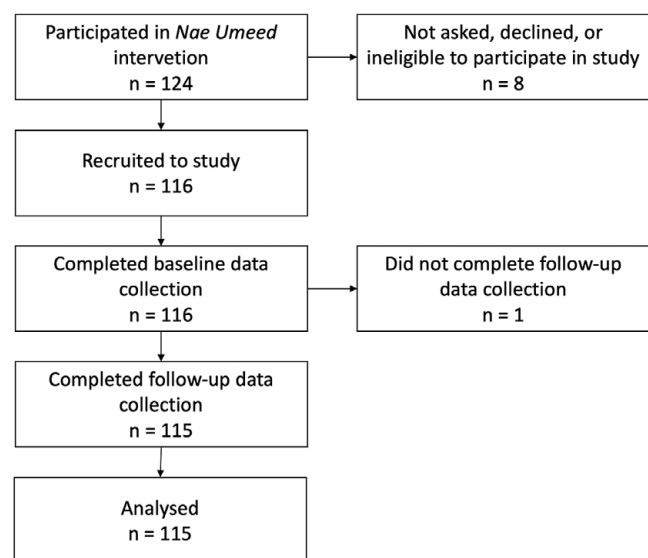

**Figure 1.** Flowchart showing eligibility, recruitment, follow-up and inclusion in analysis.

Burans staff, including members of gender-based violence support groups).

Pre- and post-intervention outcome scores are summarised in Table 3. Significant mean improvements between the pre- and post- assessments were observed for both primary (PHQ-9: 5.7 points (95% CI: 4.6–6.7), GHQ-12: 7.5 points (95% CI: 6.1–8.8)) and secondary outcomes of social participation (P-scale: 9.8 points (95% CI: 7.3–12.3)). In subgroup analyses, statistically significant improvements were observed for both caregiver and non-caregiver groups.

In the multivariable linear regression models, worse (higher) pre-intervention scores were strongly associated with larger improvements in all outcomes (Table 4), meaning those with more room to benefit, improved more. Widowed or separated participant marital status (compared with married) was associated with significantly less improvement of the GHQ-12 but not on other measures. No consistent effects were detected across other predictor variables.

### Discussion

Over the period of this study, *Nae Umeed* participants self-reported improved general well-being, greater social participation and fewer depressive symptoms using validated psychometric scales. Improvements were noted irrespective of caregiver status. Participants from vulnerable or marginalised groups such as women, people of disadvantaged caste and people with lower levels of education were well represented in the intervention, and there was no strong evidence suggesting these socio-demographic markers of disadvantage limited their capacity for benefit.

These findings are broadly consistent with the limited existing evidence for effectiveness of community health worker–delivered interventions in LMICs for mental health care and prevention (Cochrane Effective Practice and Organisation of Care Group et al., 2021; van Ginneken et al., 2021). Specific evidence for effectiveness of group caregiver interventions exists for high-income settings (Sörensen et al., 2002; Cheng and Zhang, 2020; Hovadick et al., 2021; McLoughlin, 2022), but is limited in the

**Table 2.** Demographic characteristics of 115 study participants included in analysis by caregiver status (number of participants and percent of sample, unless otherwise specified)

| | Non-caregivers (*N* = 92) | Caregivers (*N* = 23) | Overall (*N* = 115) |
|---|---|---|---|
| **Age (years)** | | | |
| Median (range) | 35 (18, 70) | 35 (14, 60) | 35 (14, 70) |
| **Gender** | | | |
| Male | 6 (6.5%) | 3 (13.0%) | 9 (7.8%) |
| Female | 86 (93.5%) | 20 (87.0%) | 106 (92.2%) |
| **Marital status** | | | |
| Married | 71 (77.2%) | 16 (69.6%) | 87 (75.7%) |
| Widowed | 12 (13.0%) | 1 (4.3%) | 13 (11.3%) |
| Separated | 2 (2.2%) | 0 (0%) | 2 (1.7%) |
| Unmarried | 7 (7.6%) | 6 (26.1%) | 13 (11.3%) |
| **Caste** | | | |
| General | 35 (38.0%) | 7 (30.4%) | 42 (36.5%) |
| Other backwards class | 22 (23.9%) | 8 (34.8%) | 30 (26.1%) |
| Scheduled caste/Scheduled tribe | 21 (22.8%) | 6 (26.1%) | 27 (23.5%) |
| N/A | 14 (15.2%) | 2 (8.7%) | 16 (13.9%) |
| **Religion** | | | |
| Hindu | 57 (62.0%) | 12 (52.2%) | 69 (60.0%) |
| Muslim | 34 (37.0%) | 11 (47.8%) | 45 (39.1%) |
| Sikh | 1 (1.1%) | 0 (0%) | 1 (0.9%) |
| **Years of education** | | | |
| 0 | 60 (65.2%) | 11 (47.8%) | 71 (61.7%) |
| 1–5 | 12 (13.0%) | 4 (17.4%) | 16 (13.9%) |
| 6–10 | 18 (19.6%) | 5 (21.7%) | 23 (20.0%) |
| >10 | 2 (2.2%) | 3 (13.0%) | 5 (4.3%) |
| **Housing type[a]** | | | |
| Kaccha | 37 (40.2%) | 5 (21.7%) | 42 (36.5%) |
| Semi-pucca | 22 (23.9%) | 7 (30.4%) | 29 (25.2%) |
| Pucca | 33 (35.9%) | 11 (47.8%) | 44 (38.3%) |
| **Household structure** | | | |
| Joint family | 15 (16.3%) | 7 (30.4%) | 22 (19.1%) |
| Nuclear family | 77 (83.7%) | 16 (69.6%) | 93 (80.9%) |

[a]Pucca, permanent houses constructed of conventional modern building materials; kaccha: semi-permanent houses made of mud, unfired bricks, grasses and makeshift materials; semi-pucca: a combination.

South Asian context. Studies of group interventions for family caregivers of persons with schizophrenia (Sims et al., 2022) and dementia (Lamech et al., 2020; Stoner et al., 2022) in India have been described but did not include measures of effectiveness. In the only published study (to our knowledge) quantitatively assessing effectiveness of a group caregiver intervention in India, a support group intervention was associated with increased family system strength scores in rural caregivers of stroke patients (Malini, 2015).

Other interventions to improve caregiver well-being in India have had mixed success. Facility-based educational interventions, predominantly for caregivers of people with psychosocial disability, have some evidence for effectiveness (Das et al., 2006; Kulhara et al.,

2009; Chakraborty et al., 2014; Singh et al., 2021). A home-care support intervention was associated with improvement in mental health of caregivers of people with dementia in a randomised controlled trial (RCT) in Goa (Dias et al., 2008). In another RCT, a multicomponent community care intervention had no significant effects on 'burden' reported by caregivers of people with schizophrenia (Chatterjee et al., 2014). An attempt to trial an online intervention for dementia caregivers suffered from low retention (Baruah et al., 2021). The current study adds to limited evidence for community-based group interventions LMICs, which may represent an efficient strategy to address mental health disparities in resource-limited settings (Hinton et al., 2019).

**Table 3.** Participant outcome scores before and after participating in the *Nae Umeed* intervention, overall and by caregiver status

| | Mean score | | | |
|---|---|---|---|---|
| Outcome | Pre-intervention | Post-intervention | Mean difference (95% CI) | *p*-value |
| PHQ-9[a] | | | | |
| Overall | 11.4 | 5.7 | 5.7 (4.6–6.7) | <0.001 |
| Caregivers | 7.8 | 4.2 | 3.7 (2.1–5.2) | <0.001 |
| Non-caregivers | 12.3 | 6.1 | 6.2 (4.9–7.4) | <0.001 |
| GHQ-12[b] | | | | |
| Overall | 15.2 | 7.8 | 7.5 (6.1–8.8) | <0.001 |
| Caregivers | 11.0 | 6.7 | 4.3 (2.7–5.9) | <0.001 |
| Non-caregivers | 16.3 | 8.1 | 8.2 (6.6–9.8) | <0.001 |
| P-scale[c] | | | | |
| Overall | 15.2 | 5.3 | 9.8 (7.3–12.3) | <0.001 |
| Caregivers | 11.1 | 4.2 | 7.0 (1.7–12.2) | 0.012 |
| Non-caregivers | 16.2 | 5.6 | 10.6 (7.7–13.4) | <0.001 |

[a]patient health questionnaire-9.
[b]short general health questionnaire.
[c]participation scale.

In the current study, improvements were observed regardless of caregiver status, suggesting that *Nae Umeed* may operate via mechanisms not specific to caregivers. In fact, larger improvements were observed among non-caregivers than caregivers. This finding may be explained by lower (better) pre-intervention outcome scores across outcome scales among caregivers versus non-caregivers recruited to this study (leaving less room for improvement), rather than reduced effectiveness due to caregiver status. This supposition is supported by results of the multivariable analyses showing negligible effects of caregiver status after adjustment for pre-intervention score and demographic variables. In North India, people with poor mental health struggle with social exclusion, finances and lack of access to care (Mathias et al., 2015a, 2018). *Nae Umeed* includes content on self-care, managing stress, psychoeducation and financial literacy, as well as offering a potential mechanism to strengthen social inclusion through peer support. These aspects of the intervention may be of wider relevance to people dealing with mental health issues of a family member, or their own. The mixed nature of groups in this study likely meant that participation was less stigmatising for all participants, potentially contributing to positive outcomes.

Several factors should be considered in trialling or implementing *Nae Umeed* or similar interventions in other settings. *Nae Umeed* was designed for the setting of low-income families in rural and urban Uttarakhand and may require some adaptations for other contexts. For example, some aspects of the financial inclusion modules are specific to Indian economic settings. Caregivers elsewhere will face different sets of issues that may warrant different content or delivery. Piloting in new target settings will be necessary to inform these adaptations. The organisational context should also be carefully considered. In this study, *Nae Umeed* was implemented via a well-established platform with strong community relationships. Facilitators were community health workers with ties to communities in which they were working. These factors likely promoted recruitment and retention and possibly effectiveness.

This study is strengthened by low drop-out and integration with an existing community mental health project. The main limitation is the absence of a comparison group, meaning the attribution of outcome improvements to the intervention is not clear. The study overlapped with a decline in India's first wave of COVID-19 and the easing of associated public health restrictions, shifts which probably had independent positive effects on the mental and social well-being of participants. Social desirability bias may have also contributed to the positive outcomes at the follow-up assessment, particularly as outcomes were solicited in-person by a community health worker. Recruiters may have focussed on including those they felt were more likely to benefit from the intervention; the total number of identified eligible individuals is not available. These biases could have led to overestimation of the effectiveness of *Nae Umeed*. Outcomes were assessed within 3 weeks after completion of the intervention, and a follow-up would be required to assess how long these benefits were sustained.

Future research should focus on assessing sustained effects on caregiver mental health, as well as exploring intervention mechanisms and implementation issues. A cluster RCT with longer follow-up would provide a more confident estimate of intervention effectiveness. The findings of this study highlight the current evidence gap and provide preliminary evidence for effectiveness. Ongoing qualitative research will help tailor *Nae Umeed,* identify optimal measurable outcomes for future studies, and explore barriers and facilitators to implementation in the current setting.

## Conclusions

The findings of this study are consistent with the effectiveness of *Nae Umeed* in improving mental health and social participation in caregiver and non-caregiver participants; however, further research is required to establish the degree to which improvements can be causally attributed to the intervention. Nevertheless, the intervention was successful in reaching marginalised target groups typically

**Table 4.** Linear regression coefficients for the mutually adjusted effects of participant socio-demographic variables on a standard deviation improvement in outcome score

| Characteristic | Improvement on PHQ-9[a] | | | Improvement on GHQ-12[b] | | | Improvement on p-scale[c] | | |
|---|---|---|---|---|---|---|---|---|---|
| | Beta | 95% CI[d] | p-value | Beta | 95% CI | p-value | Beta | 95% CI | p-value |
| Baseline score | 0.77 | 0.65, 0.89 | <0.001 | 0.82 | 0.71, 0.93 | <0.001 | 0.88 | 0.78, 1.0 | <0.001 |
| Age (years) | −0.01 | −0.02, 0.01 | 0.3 | 0.01 | −0.01, 0.02 | 0.4 | −0.01 | −0.02, 0.00 | 0.2 |
| Gender | | | | | | | | | |
| Male | | | | | | | | | |
| Female | −0.08 | −0.56, 0.41 | 0.8 | 0.24 | −0.16, 0.65 | 0.2 | 0.24 | −0.15, 0.62 | 0.2 |
| Marital status | | | | | | | | | |
| Married | | | | | | | | | |
| Widowed | −0.16 | −0.55, 0.24 | 0.4 | −0.52 | −0.85, −0.19 | 0.003 | −0.01 | −0.33, 0.31 | >0.9 |
| Separated | −0.49 | −1.4, 0.39 | 0.3 | −1.0 | −1.7, −0.26 | 0.010 | 0.48 | −0.22, 1.2 | 0.2 |
| Unmarried | 0.14 | −0.34, 0.62 | 0.6 | 0.16 | −0.24, 0.56 | 0.4 | −0.30 | −0.67, 0.08 | 0.13 |
| Caste | | | | | | | | | |
| General | | | | | | | | | |
| Other backwards class | −0.15 | −0.51, 0.21 | 0.4 | 0.65 | 0.35, 1.0 | <0.001 | 0.19 | −0.09, 0.48 | 0.2 |
| Scheduled caste/Scheduled tribe | 0.00 | −0.32, 0.31 | >0.9 | 0.18 | −0.08, 0.45 | 0.2 | 0.19 | −0.06, 0.43 | 0.14 |
| N/A | −1.0 | −1.3, −0.60 | <0.001 | −0.23 | −0.54, 0.07 | 0.14 | −0.01 | −0.30, 0.28 | >0.9 |
| Religion | | | | | | | | | |
| Hindu | | | | | | | | | |
| Muslim | −0.10 | −0.43, 0.24 | 0.6 | −0.20 | −0.48, 0.09 | 0.2 | 0.08 | −0.19, 0.34 | 0.6 |
| Sikh | 1.0 | −0.23, 2.2 | 0.12 | 0.78 | −0.22, 1.8 | 0.13 | 0.34 | −0.59, 1.3 | 0.5 |
| Years of education | | | | | | | | | |
| 0 | | | | | | | | | |
| 1–5 | −0.11 | −0.46, 0.24 | 0.5 | 0.07 | −0.22, 0.36 | 0.6 | −0.07 | −0.34, 0.20 | 0.6 |
| 6–10 | −0.14 | −0.44, 0.17 | 0.4 | 0.07 | −0.19, 0.32 | 0.6 | 0.09 | −0.16, 0.33 | 0.5 |
| > 10 | −0.04 | −0.66, 0.58 | 0.9 | 0.21 | −0.31, 0.73 | 0.4 | 0.22 | −0.27, 0.70 | 0.4 |
| Housing type[e] | | | | | | | | | |
| Kaccha | | | | | | | | | |
| Semi-pucca | −0.01 | −0.32, 0.29 | >0.9 | −0.07 | −0.32, 0.19 | 0.6 | −0.01 | −0.25, 0.23 | >0.9 |
| Pucca | 0.05 | −0.25, 0.35 | 0.7 | 0.28 | 0.03, 0.54 | 0.032 | 0.05 | −0.18, 0.29 | 0.7 |
| Household structure | | | | | | | | | |
| Joint family | | | | | | | | | |
| Nuclear family | 0.24 | −0.07, 0.54 | 0.13 | 0.17 | −0.09, 0.42 | 0.2 | −0.13 | −0.37, 0.11 | 0.3 |
| Participant type | | | | | | | | | |
| Non-caregivers | | | | | | | | | |
| Caregivers | 0.05 | −0.25, 0.35 | 0.7 | −0.11 | −0.36, 0.14 | 0.4 | 0.00 | −0.23, 0.22 | >0.9 |

[a]patient health questionnaire-9.
[b]short general health questionnaire.
[c]participation scale.
[d]CI, confidence interval.
[e]Pucca, permanent houses constructed of conventional modern building materials; kaccha: semi-permanent houses made of mud, unfired bricks, grasses, and makeshift materials; semi-pucca: a combination.

not well serviced by the traditional mental health care system. Community-based group interventions are a promising but under-explored strategy for addressing mental health disparities for vulnerable populations in South Asia.

**Open peer review.** To view the open peer review materials for this article, please visit http://doi.org/10.1017/gmh.2023.38.

**Data availability statement.** The data that support the findings of this study are available from the corresponding author upon reasonable request. The

complete data are not publicly available due to their containing information that could compromise the privacy of research participants.

**Acknowledgements.** Thanks to the whole Burans team in Dehradun district for the great work implementing *Nae Umeed* and to the CHGN-Cluster for support in implementation at the peri-urban site. Appreciation to Dr. Shubha Nagesh for the interviews she conducted to help understand feasibility and acceptability of this intervention in the community. Thanks to Herbertpur Christian Hospital for their ongoing support to all implementation and evaluation work.

**Author contribution.** Conceptualisation: P.S.P., K.M.; Formal analysis: C.R. B., J.L.,P.S.P.,K.M.; Investigation: P.S.P., A.G.S.; Methodology: C.R.B., P.S.P., K. M.; Project administration: P.S.P., A.G.S.; Supervision: N.J.G., K.M.; Writing (original draft): C.R.B.; Writing (review and editing): P.S.P., N.J.G., K.M.

**Financial support.** Research costs were covered by existing programme funds.

**Competing interest.** The authors declare none.

**Ethics standard.** Approval for this project was provided by the institutional ethics committee of the Emmanuel Hospital Association (protocol number: 240).

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
