## [Reviewer Report]

Thirdly, we are a multidisciplinary, multi-country authorship team who are truly global in nature. PP and AGS are respectively managerial and social work -based mental health practitioners based in a non-profit organisation in North India; KM works and researches in community mental health in New Zealand and India, NG and CB are public health physicians working between Australia and India and JL is a physician who volunteered to support with analysis who works in New Zealand and Australia.

We believe this paper fits Global mental health well, and offers a solid account of the pilot evaluation of the effectiveness of a novel caregiver group intervention.

Thank you for your consideration of this paper for publication,

Kaaren Mathias – on behalf of the author team

---

## [Reviewer Report]

The study looks very promising in order to address care and improve mental health in settings where services are scarce. The attention to caregivers is complemented by the non caregiver’s group, that showed even better results. The low drop-in rate shows that the intervention is accepted and sustainable. The article should be complemented by another study that describes more in depth the content of the program, in order to facilitate its dissemination. Immediate comparison arises with psycho-education provided to family members in HIC in order to support them for the high burden they perceive. Nonetheless, its seems that this program is more focused on social participation and well-being rather than on coping with illness of a relative and related information.

The lack of a control group is a limit (recognized by the authors), as well as the short-term evaluation after its completion. It will be important to understand if the outcomes will be maintained over a longer period of time.

Statistical analysis looks sufficient to the scope of the study.

---

## [Reviewer Report]

Congratulations on completing this much-needed piece of work to develop and test interventions for caregivers in India. I have a few suggestions and queries that can help to improve the quality of the manuscript.

1. Page 4 Lines 70-73: The gender relations in India mean that most caregivers are female (Janardhana et al. 2015, Chakrabarti 2016) and may experience greater challenges in sustaining caregiving due to systematic disadvantage (World Economic Forum 2021), leading to (Mathias et al. 2019). This sentence appears incomplete.

2. While the authors mention that including non-caregivers helped in improving social participation and reducing stigma, I think this seriously limits the ability of the study to measure improvements in caregivers. I believe that Nae Umeed was developed specifically for caregivers. Only a fifth of the participants is caregivers.

3. Ideally to study the effectiveness of an intervention, a blinded randomised controlled study design should be used. In this case, we cannot say the authors have measured the effectiveness. They have found a positive trend towards improvement.

4. Is informed verbal consent adequate? Is there a record of the consent forms available?

5. Among the participants who were caregivers, it would be helpful for the readers to know, who they were providing care for and what the nature of the illnesses was.

6. Among 75 participants with psychosocial disability, what was the nature of the psychosocial disability? Did they have a formal diagnosis?

7. The authors should elaborate on what they mean by “others” about the 17 participants.

8. The authors report that a larger improvement was noted among non-caregivers. The intervention clearly is not specific to caregivers. For those with poor mental health status, how many were receiving formal support, including medicines?

---

## [Reviewer Report]

1. This is a good attempt to understand the impact of a structured program even though it would have been better had the study been a RCT design rather than pre post design. Of course the authors are acknowledging this and recommending this as a future design

2. Because this is not an RCT, the authors should reconsider calling this an effectiveness study.

3. Since the study participants consisted of only 20% caregivers, the title of the study is misleading. It would be advisable to consider changing the title.

3. As there were only 23 caregivers in the study, the study cannot claim that this is a promising approach for caregivers as such

4. The duration of caregiving and types of disorders in the affected family members will be useful to be provided in the sample description of the caregivers.

5. The consort diagram does not give the number of people approached for the study and the proportion of caregivers and non caregivers who agreed to participate as this will indicate the generalisability of the findings.

6. As participants included both caregivers and people with psychosocial disability, the authors should clarify if the mixed group included people from the same families. This will have an impact on the outcome too

7. It is not clear from the description of the procedure, how they determined the sample size of 100 to 130. It is not sufficient to simply say it is a pragmatic sample size.

8. It will be useful for the readers to know what were the mean number of sessions attended by the participants.14 sessions appears a lot of commitment for participants

9. The authors should describe how many community health workers delivered how many groups and how was the fidelity to the intervention checked and confirmed.

10. The authors have stated in line 218-219 that worse pre-intervention scores were associated more with more improvement and the non caregiver group had worse baseline scores. The authors should discuss the impact of this on the overall outcome of the study.

11. In Line 232 the authors report that their findings are consistent high income country findings. However with such a small sample of caregivers in this study, I would be concerned with this statement.

12. In the conclusion in line 297, the authors speak about peer group intervention. The community health workers are form the same community but not peers in the working efinition.

---

## [Reviewer Report]

This is an interesting manuscript that describes a locally developed caregiver intervention and its effectiveness in improving mental health and social participation among community participants in Dehradun, Uttarakhand, India. Overall, the manuscript is easy to read, well written, and highlights the significance of a community-based group intervention program and has important implications for public mental health. However, there are certain queries and issues in the manuscript that need to be addressed. Some of the queries, concerns/comments are mentioned below:

Title: Though primarily developed for caregivers of persons with disability, in the current study only 20% of the participants were actual caregivers. In this context, how do the authors justify the intervention as a “caregiver” group intervention

Impact Statement and Abstract:

Both in the impact statement and the abstract, it should be mentioned that the participants were largely non-caregivers, as it is currently not clear.

Background:

Line 71-73 seems incomplete and needs to be revised.

Line 79-80: caregiver burden and mental health impact have been studied extensively in caregivers of persons with dementia in India - relevant citations can be included.

Methods:

Has the development process of Nae Ummed been described elsewhere? If yes, it would be useful to provide a reference for the same in the manuscript

Though the reference for the structured curriculum has been provided by the authors, it would be helpful if a brief description/summary of the sessions/modules can be provided (in a table or in text). It would also be helpful to understand how the topics were adapted for non-caregivers as the modules were primarily developed for caregivers.

Line 141-143: in the results it is mentioned that 62% of the participants had no formal education. So, were any alternatives provided for the pamphlets on self-care? Or were the pamphlets with visual descriptions?

Line 144-147: who provided training to the CHWs? And what were their credentials?

Line 157-158: operational definition of people with disability can be provided with a mention of the types of disabilities included.

Line 160: Were there any specific criteria for caregivers with respect to duration of caregiving, caregiving role, etc.?

Line 166: the process of obtaining verbal informed consent can be elaborated for clarity for the readers.

Data collection: how many health workers were involved in the study – as group facilitators and in the data collection and recruitment process?

Trial registration: why was the study registered in the Australia New Zealand Clinical Trials Registry when it was conducted in India?

Results:

In the socio-demographic profile, the addition of the employment status and average income/economic status of the participants would provide a holistic description of the target population.

Line 211: who are these “others” and were they clubbed with non-CGs in the analysis?

Line 212: Was the data checked for normality?

Discussion:

The authors discuss the results primarily in the context of caregiver support group interventions, however, as the participants were mostly non-caregivers, the results can be discussed in terms of community mental health interventions.

The strengths of the study/methods (viz., low dropout rate, etc.) can be emphasized

Was there any qualitative feedback collected from the participants? During the sessions, did the participants who were caregivers have questions about how to manage illness-specific issues of the persons they were caring for and how were these handled?

---

## [Reviewer Report]

COVER LETTER

Dear Editors of Global mental health,

We have made revisions to this paper based on the thoughtful feedback from all four reviewers. We believe this paper is now in a strong shape for publication. We underline what we stated at the time of submission that this is an important contribution to the evidence of global mental health for the following reasons:

First, this paper profiles the pilot implementation of a new intervention developed for North India, that seeks to strengthen the mental health and social participation of people who are mentally distressed in urban North India including carers. This acknowledges that the centrality of carers in mental health care systems in low income settings such as urban North India. This study evaluates a pragmatic intervention seeking to address the mental health needs of caregivers. As an intervention developed and implemented by a civil society organisation, this evaluates a pilot programme with a pragmatic and uncontrolled research design. 

Secondly, an important strength of this study, is that is demonstrates significant improvements in pre-post measures for mental health and social participation for people with mental distress in a real-world setting using pragmatic implementation. The low-budget implementation by a civil society organisation using a group platform during the period of the COVID aftermath increases the scalability, generalisability and relevance of the positive findings.

Thirdly, we are a multidisciplinary, multi-country authorship team who are truly global in nature. PP and AGS are respectively managerial and social work -based mental health practitioners based in a non-profit organisation in North India; KM works and researches in community mental health in New Zealand and India, NG and CB are public health physicians working between Australia and India and JL is a physician who volunteered to support with analysis who works in New Zealand and Australia.

We believe this paper fits Global mental health well, and offers a solid account of the pilot evaluation of the effectiveness of a novel caregiver group intervention.

Thank you for your consideration of this paper for publication,

Kaaren Mathias – on behalf of the author team

---

## [Reviewer Report]

Despite some weaknesses in the methods, I think this is an important study that needs to be published.

---

## [Reviewer Report]

The authors have satisfactorily addressed the concerns and queries I had. They have incorporated the suggestions given.